# The Mitochondrial Permeability Transition: Nexus of Aging, Disease and Longevity

**DOI:** 10.3390/cells10010079

**Published:** 2021-01-06

**Authors:** Hagai Rottenberg, Jan B. Hoek

**Affiliations:** 1New Hope Biomedical R&D, 23 W. Bridge street, New Hope, PA 18938, USA; 2MitoCare Center, Department of Anatomy, Pathology and Cell Biology, Thomas Jefferson University, Philadelphia, PA 19107, USA; JanHoek@jefferson.edu

**Keywords:** mitochondrial permeability transition, aging, longevity, aging-driven degenerative disease, reactive oxygen species, mitophagy, autophagy, Parkinson’s disease

## Abstract

The activity of the mitochondrial permeability transition pore, mPTP, a highly regulated multi-component mega-channel, is enhanced in aging and in aging-driven degenerative diseases. mPTP activity accelerates aging by releasing large amounts of cell-damaging reactive oxygen species, Ca^2+^ and NAD^+^. The various pathways that control the channel activity, directly or indirectly, can therefore either inhibit or accelerate aging or retard or enhance the progression of aging-driven degenerative diseases and determine lifespan and healthspan. Autophagy, a catabolic process that removes and digests damaged proteins and organelles, protects the cell against aging and disease. However, the protective effect of autophagy depends on mTORC2/SKG1 inhibition of mPTP. Autophagy is inhibited in aging cells. Mitophagy, a specialized form of autophagy, which retards aging by removing mitochondrial fragments with activated mPTP, is also inhibited in aging cells, and this inhibition leads to increased mPTP activation, which is a major contributor to neurodegenerative diseases, such as Alzheimer’s and Parkinson’s diseases. The increased activity of mPTP in aging turns autophagy/mitophagy into a destructive process leading to cell aging and death. Several drugs and lifestyle modifications that enhance healthspan and lifespan enhance autophagy and inhibit the activation of mPTP. Therefore, elucidating the intricate connections between pathways that activate and inhibit mPTP, in the context of aging and degenerative diseases, could enhance the discovery of new drugs and lifestyle modifications that slow aging and degenerative disease.

## 1. Introduction: The Mitochondrial Permeability Transition Pore

The mitochondrial permeability transition pore (mPTP) is a mitochondrial inner membrane multicomponent mega-channel, with variable conductance (up to 1.5 nS) that is activated by calcium, oxidative stress and membrane depolarization [1,2]. The channel exhibits several conductance states with variable duration. When activated, protons flow into the matrix, while calcium, superoxide, hydrogen peroxide and other ions flow out of the matrix and the mitochondrial protonmotive force (∆Ѱ + ∆pH) collapses, thus inhibiting oxidative phosphorylation. The high conductance pore, when fully open, also allows the passage of large solutes, with MW up to 1.5 KDa, and the outflow of respiratory substrates from the matrix, which are normally held at a high concentration in the matrix by the protonmotive force, inhibits electron transport, while flooding the matrix with cytosolic solutes leads to swelling of the mitochondrial matrix and eventually rupture of the outer mitochondrial membrane. The inhibition of oxidative phosphorylation depletes cellular ATP and therefore extensive and prolonged activation of mPTP may lead to cell death by necrosis; in addition, the rupture of the mitochondrial outer membrane releases proapoptotic proteins, including cytochrome c, AIF and endonuclease, thereby inducing cell death by apoptosis or similar processes [3,4,5,6,7,8,9,10]. Oxidative stress-dependent cell death largely depends on the activation of mPTP. Lower conductance states with a short duration of partial activation may release only small solutes and ions, such as superoxide, hydrogen peroxide and calcium, and this release may play an important role in H_2_O_2_ and Ca^2+^ signaling [11,12,13]. For example, mPTP-mediated superoxide flashes regulate cortical neural progenitor differentiation [12]. Moderate activation of mPTP may be insufficient to cause cell death but may result in damage to both mitochondrial and cellular proteins, lipids and DNA and thus accelerate cell aging [14].

The exact composition of mPTP is still not fully resolved. There are several mitochondrial proteins that were shown to participate in the channel activity, such as cyclophilin D (CypD), adenine nucleotide translocase (ANT), ATP synthase, the outer membrane voltage-dependent anion channel (VDAC), the phosphate carrier (PiC) and SPG-7. Because CyPD is not a transmembrane protein it cannot form a channel on its own, but it binds to pore-forming protein(s) and regulates their channel activity, as is evident from the inhibitory effect of its ligand cyclosporin A [15,16,17]. Reconstitution of ANT, VDAC, PiC, ATP synthase and subunit c of ATP synthase in liposomes, or in planar phospholipid membranes, showed that a channel can be formed that exhibits, at least in part, the properties of mPTP. However, genetic ablation studies of each one of these candidate proteins showed that none of these proteins is essential for mPTP activity, although the residual activity was always somewhat different from mPTP activity in wt mitochondria [1,18]. The fact that mPTP can be induced in cells depleted of mtDNA suggests that the composition and activity of the pore does not require mtDNA-coded proteins [19]. The inescapable conclusion from these studies is that some combination of these proteins is necessary to fully exhibit the properties of mPTP [1,18,20,21]. Because experiments with ATP synthase and ANT provided the strongest evidence for participation in the mPTP channel, it was recently suggested that both ATP synthase and ANT are required for the formation of a fully functional mPTP channel, although the nature of this composite channel remains elusive [1,20,21]. Moreover, the contribution of several other proteins to mPTP activity is still unresolved. In particular, the outer membrane transporter VDAC has been shown, by several studies, to control mPTP activity [6,22,23,24,25,26,27]. It was previously shown that VDAC interacts with ANT to form a channel [28] and that the reconstitution of a complex of ANT/VDAC/CypD exhibits mPTP-like activities [29]. Moreover, VDAC was shown to lock ANT in the C conformation [30], which is known to activate mPTP [2,31]. A recent study demonstrated the co-immunoprecipitation of SGK1/VDAC1 with nearly all other protein candidates for the mPTP channel, i.e., ANT1, ANT3, two peptides of ATP synthase (OSCP and subunit delta), SPG-7 and PiC, but no other mitochondrial proteins [27]. Furthermore, the effects of VDAC1 accumulation on mPTP activation, autophagy and lifespan were dependent on ANT1 [27], suggesting that the activation of mPTP by VDAC1 is mediated by the VDAC/ANT complex.

## 2. Aging and Enhanced mPTP Activity

Aging is a process of gradual accumulation of damage to cellular proteins, lipids, DNA and cell organelles, leading to cellular, organellar and organ dysfunctions, resulting in aging-driven diseases, cell death and finally organism death [32,33,34]. It is now recognized that mitochondrial dysfunction is a major contributor to aging and aging-driven degenerative disease, such as diabetes, heart diseases, cancer, Alzheimer’s disease and Parkinson’s disease [35,36,37,38,39,40]. Mitochondrial dysfunction in aging is often manifested as the excess production of mROS, calcium overloading, and membrane depolarization. Since these dysfunctions are known to activate mPTP [2], it can be expected that mPTP activity will be enhanced in dysfunctional mitochondria in aging. Indeed, direct evidence for enhanced mPTP activation in aging and neurodegenerative disease is extensive. This evidence has been reviewed frequently and will not be described in detail in this review [41,42,43,44,45,46].

It has been recognized for a long time that mitochondria are the major source of ROS in the cell and that oxidative damage to phospholipids, proteins, mtDNA and nuclear DNA in aging results from the excess production of mROS [47,48,49,50,51,52]. Mitochondrial metabolic reactions continuously generate superoxide from several sources, including the citric acid cycle enzymes and electron transport enzymes with variable rates that depend on the metabolic pathways, on the redox state of key components and on the mitochondrial membrane potential [53,54,55,56,57,58]. However, the mitochondrial matrix also contains a robust system that converts superoxide to H_2_O_2_ (SOD2), and several peroxidases that regulate the level of H_2_O_2_ in the matrix [59]. While the mitochondrial inner membrane is impermeable to both superoxide and H_2_O_2_ [60], a water channel, aquaporin 8 (AQP8), allows the diffusion of H_2_O_2_ to the outer compartment [61] and, eventually, through VDAC to the cytoplasm. That flow of H_2_O_2_ from the mitochondria to the cytoplasm largely controls the redox balance in the cytoplasm. A metabolically induced change in H_2_O_2_ flow from the mitochondria will change the redox balance in the cell, which exerts its effects on metabolic pathways, through its impact on -SH residues on critical proteins [62]. In recent years, it has become apparent that a moderate increase in mROS production is actually beneficial to the cell as it serves as a signal to the nucleus to activate a number of mechanisms that protect the cell, and particularly the mitochondria, from the destructive effects of mROS [63,64,65,66,67,68]. Partial, short duration openings of mPTP can generate a pulse of H_2_O_2_, Ca^2+^ and superoxide that may serve as signals for several physiological processes [11,12,13,69,70]. What determines whether the channel opens for a short or long duration is not entirely clear [71]. However, a full and longer lasting opening of the pore can generate a large excess of superoxide and H_2_O_2_ release that can overpower the mitochondrial and cellular antioxidant systems and cause damage to membrane phospholipids, enzymes, transporters and most importantly DNA [72]. The mROS released by the activation of mPTP at one mitochondrial site may activate mPTP at an adjacent site and this second opening can then trigger opening at other sites, creating a propagating wave of mPTP opening across the cell [54]. The opening of mPTP, in addition to the fast release of the mROS content of the matrix, induces further production of superoxide while the mPTP remains open. It appears that the inhibition of oxidative phosphorylation rather than inhibiting superoxide production actually stimulates the production of mROS at specific sites [73,74,75,76]. When mPTP is fully activated, and this activation is propagated throughout the cell, the inevitable outcome is cell death, as described above, unless the process can be stopped or reversed before the cell death processes progress. The opening of mPTP can be reversed readily if the pore is only partially open since Ca^2+^, which is required to keep the pore open, is lost quickly; in addition, proton flow into the matrix lowers the pH, which inhibits the channel [2], and membrane potential is restored to close the pore. However, when the pore is fully open, the mitochondrial respiratory substrates that are at much higher concentrations in intact mitochondria would be lost during the opening of mPTP, and electron transport cannot recover unless the substrates are taken up by the mitochondria, a process that itself depends on the proton motive force. Nevertheless, ATP, which is normally at a higher concentration in the cytoplasm than in the matrix, could flow through the pore into the mitochondria and reverse the ATPase, which would restore the proton motive force [77,78,79]. This would close the pore and drive the re-accumulation of respiratory substrates, restarting oxidative phosphorylation, and allow the mitochondria to recover. Thus, unless the mPTP activation has propagated massively, and cellular ATP is already depleted, the mitochondrion can recover after full activation of mPTP. However, the oxidative damage to mitochondrial and cellular proteins, lipids and DNA, that was done by the extended production of mROS, cannot be fully erased, and cell aging will progress. Another critical mechanism that can stop the propagation of mPTP opening in the cell is the removal of mitochondria with activated mPTP by mitophagy, a process that protects the cell from progression to apoptosis, reduces oxidative damage and retards aging (see below). An additional deleterious outcome of extended mPTP opening is the loss of NAD^+^. NAD^+^ is the substrate of both the NAD^+^-dependent deacetylases, sirtuins and PARP1, that coordinate DNA repair. The loss of NAD^+^ from the mitochondrial matrix inhibits the mitochondrial sirtuins (sirt3, sirt4, sirt5), resulting in increased mROS generation [80], and also enhanced activity of mPTP because CypD is inhibited by deacetylation by sirt3 [81,82]. Therefore, even when the mPTP opening is reversed, and the mitochondria recover, the loss of NAD^+^ would leave the mitochondria more susceptible to a second opening of the pore [83]. Moreover, the NAD^+^ that exits the matrix into the outer compartment is hydrolyzed by CD38, depleting the cellular pool of NAD^+^ [3,84], thus also inhibiting cytoplasmic sirtuins (e.g., sirt1). In addition, the release of mROS by mPTP opening activates PARP1 which further depletes cellular NAD^+^ [85,86]. It is now recognized that one of the major causes of aging and degenerative disease is the depletion of NAD^+^ in aging cells [87,88,89,90,91,92]. CD38 expression increases with age, which further enhances the destructive effects of mPTP opening in aging [93]. The fact that the opening of mPTP further enhances the production of damaging mROS and leads to the cellular depletion of NAD^+^ led us to suggest that mPTP activity is critical for the progression of aging [14].

The oxidative damage to cell proteins, lipids and most importantly nuclear DNA is now believed to be a critical element of the aging process [94,95,96,97,98,99,100]. Oxidative damage to nuclear DNA elicits the DNA damage response that induces both proapoptotic pathways and protection pathways [101]. Several protection pathways depend on induction by mROS of PARP1, which repairs damaged DNA in an NAD^+^-dependent manner [102], and on the induction of NAD^+^-dependent deacetylases, the sirtuins [103,104]. Of critical importance is sirt1, which deacetylates a number of critical proteins [88,105,106]. Similar to deacetylases, histone demethylases also contribute to stress-induced protection [107]. In addition to inducing protection of nuclear DNA from oxidative damage, mROS initiate signals that activate several pathways that protect mitochondria from oxidative stress. These pathways slow aging, inhibit cell death and may result in lifespan extension [67,108,109]. The mitochondrial sirtuins (sirt3–5), and particularly sirt3, are critical in the protection of mitochondria [80,110,111]. Other pathways that protect the mitochondria are the mitochondrial unfolded protein response, UPR(mt), which enhances mitochondrial homeostasis by enhancing the expression of mitochondrial chaperones [112,113,114,115,116], and the nrf2 antioxidant response, which protects against mROS-induced mitochondrial damage and cell death [117,118,119,120]. The induction of PGC1alpha, which initiates mitochondrial biosynthesis, is also important in displacing dysfunctional mitochondria [121], and is the main mechanism by which physical exercise delays aging and protects from aging-related degenerative diseases (see below). Another important pathway that is now recognized as playing a major role in delaying aging and degenerative disease is autophagy and its mitochondrion-specific form, mitophagy (discussed below). All the protection pathways inhibit mPTP activity either directly, by regulating the expression or by posttranslational modification of mPTP components, or indirectly, by inhibiting mROS production or calcium overloading, or by eliminating damaged mitochondria by mitophagy, or by inducing the biosynthesis of new mitochondria. In contrast, several proapoptotic pathways (e.g., P53, p66Shc) enhance mPTP activity [122,123,124,125]. Figure 1 shows the major pathways that enhance or inhibit the activation of mPTP in aging.

## 3. Autophagy, Aging and mPTP

Autophagy catabolizes damaged cellular components to protect cells against stress and maintain homeostasis [126,127]. There are three types of autophagy: microautophagy, in which damaged cellular components are directly sequestered by the lysosomes for degradation, chaperone-mediated autophagy, in which specific motif-containing cargo proteins are delivered to the lysosome by chaperone complexes, and macroautophagy, in which damaged cytosolic components, including organelles, are sequestered into double-membrane autophagosomes that fuse with the lysosome. The macroautophagy process is mediated by a large number of autophagy-related proteins (ATG) specific to the various steps of the process (e.g., initiation, formation of a phagophore, cargo sequestration, fusion of autophagosome with lysosome and degradation of the cargo in the autolysosome). The process is regulated by the nutrient sensors mTOR and AMP-activated kinase (AMPK), both of which phosphorylate another kinase, ULK1, which initiates autophagy.

Because autophagy can remove damaged cell components that are associated with aging, this process, and particularly macrophagy, plays an important role in retarding aging and aging-related disease [128,129,130,131,132,133,134]. A selective form of macroauthophagy is mitophagy, which specifically removes damaged mitochondria and thus protects the cell from the deleterious effects of dysfunctional mitochondria, and specifically mitochondria with activated mPTP (see below). Other selective forms of autophagy that may also play a role in aging and aging-related disease are: lipophagy, which removes aberrant lipids, aggrephagy, which removes protein aggregates, and lysophagy, which removes damaged lysosomes.

It has been shown in model organisms (yeast, Drosophila, Caenorhabditis elegans and mice) that many paradigms of life extension depend on autophagy [34,130,135,136]. These paradigms include: dietary restriction, mTOR inhibition, reduced insulin/IGF1 signaling, increased AMPK activity, reduced mitochondrial respiration and reduced TGFb/activin signaling. Autophagy capacity decreases with age. In model animals, as well as humans, the expression and activity of autophagy genes is reduced with age in various tissues, resulting in the accumulation of intermediates of the process, indicating defective autophagy. The overexpression of specific autophagy genes leads to life extension, while the loss of function of autophagy genes is often associated with age-dependent degenerative diseases [131,137,138,139].

AMPK, the major activator of autophagy, is activated by AMP and is very sensitive to any modulation of the AMP/ADP ratio [140]. An increase in this ratio indicates a reduction of cellular ATP concentration, signaling a lack of nutrients or other stresses and forcing a shift in metabolism from anabolic metabolism to catabolic metabolism by activating AMPK. AMPK phosphorylates ULK1, thereby initiating autophagy [141]. Other activators of AMPK are a reduction in glucose concentration and, most importantly in the context of aging and disease, elevated levels of ROS [142,143,144]. AMPK phosphorylates many other key enzymes within the autophagy pathways, including key enzymes of mitophagy [145,146,147].

Another nutrient-sensing pathway that controls autophagy and thus lifespan is the mTOR pathway [148,149,150,151,152]. There are two branches in the mTOR pathway, mTORC1 and mTORC2. mTORC1 regulates cell growth and metabolism and negatively regulates autophagy. A lack of nutrients or other stresses inhibit mTORC1 and activate autophagy by enabling the phosphorylation of ULK1. mTORC2 controls cell proliferation and survival by the phosphorylation of several protein kinases, including AKT and SGK1.

While autophagy is implicated in several paradigms of life extension and is generally considered an antiaging mechanism, autophagy is also a well-defined mechanism of cell death, autophagy-dependent cell death, ADCD [153,154,155]. It has been shown recently that what determines whether autophagy is a cell protective mechanism or a cell destructive mechanism is the activation state of mPTP: when mPTP is inhibited, autophagy is protective, while overactivation of mPTP turns autophagy into a destructive process [27]. Under normal conditions, when mTORC2 phosphorylates SGK1, SGK1 phosphorylates VDAC1 at a specific site, and that phosphorylation tags VDAC1 for ubiquitination and proteasomal degradation, thereby inhibiting mPTP. In C. elegans cells, genetic interference with this process results in the accumulation of VDAC1 on the outer mitochondrial membrane, and in the ANT1-dependent activation of mPTP, leading to mitochondrial fragmentation, and a shorter lifespan. Genetic inhibition of autophagy in these cells restored the normal lifespan. In addition, genetic or pharmacological inhibition of mPTP increased the lifespan in these mutants. In SGK1-knockout mouse hepatocytes, the VDAC1 level was elevated, mPTP activity was increased, I/R susceptibility was increased and this effect was reversed by cyclosporin A. In several long-lived C. elegans models, such as calorie restriction or electron transport protein dysfunction, which are known to depend on autophagy for life extension, the stimulation of mPTP by VDAC1 overexpression abrogated the autophagy-dependent life extension. These results strongly suggest that all the antiaging effects of autophagy are contingent on the inhibition of mPTP activity, supporting the hypothesis that mPTP activity is critical for the progression of aging [14]. The results of Zhou et al. [27] were corroborated by recent studies [156,157] that similarly showed that the inactivation of mTORC2 and SGK1 in C. elegans enhanced autophagic degradation of mitochondria (mitophagy), which led to developmental and reproductive defects, and was associated with increased release of mitochondria-derived ROS (most probably resulting from the increased activation of mPTP, see below).

## 4. Mitophagy, Aging, mPTP and Parkinson’s Disease

Understanding the critical role of mitophagy and mPTP in the progression of age-driven neurodegenerative diseases has progressed greatly in recent years, particularly in relation to the most common neuronal degenerative diseases: Alzheimer’s and Parkinson’s diseases [158,159,160,161,162,163,164,165,166]. The role of mPTP in Alzheimer’s disease is discussed in this series by Heng Du and will not be discussed in this review. To understand the role of mPTP in Parkinson’s disease, we need to understand the relationship between mitophagy and mPTP.

Mitophagy is a specialized form of autophagy in which damaged mitochondria are tagged for removal by autophagy [167,168]. There are apparently several pathways to mitophagy [169,170], but the most important and the better understood one is the canonical PINK1/PARKIN pathway [171,172,173,174,175,176]. In this pathway, PTEN-induced kinase 1 (PINK1), a serin/threonine kinase, accumulates on the mitochondrial outer membrane (MOM) surface of depolarized and oxidatively stressed mitochondria [177,178]. It recruits the E3 ubiquitin protein ligase PARKIN to MOM, where it participates in the ubiquitination of mitochondrial proteins, marking the mitochondria for delivery to autophagosomes that are taken up by lysosomes. The ubiquitination is not limited to OMM proteins, as the inner membrane protein prohibitin 2 is also ubiquitinated and this process is critical for mitophagy [179]. Additionally, Nip3-like protein X (NIX) can mediate mitophagy independent of the PINK1/PARKIN pathway [169]. Mitophagy is intricately linked to mitochondrial dynamics [180,181,182]. In all cells, mitochondria undergo a dynamic cycle of fission and fusion [181,183,184,185]. In most cells, and particularly in neurons, the fused mitochondria consist of long tubular filaments forming an extended network, and there is a continuous process of fission, that breaks the elongated mitochondria into fragments, and fusion, that fuses these fragments back into tubular filaments. Mostly, this process serves to reconfigure the mitochondrial network according to cellular demand. However, damaged mitochondrial fragments are tagged for autophagy, mostly by the PINK1/PARKIN pathway [186], while the undamaged fragments, as well as newly synthesized mitochondria, are fused into elongated tubular filaments. Recent evidence suggests that the fission process is accelerated by aging [187], while fusion is inhibited [188], thereby increasing mitochondrial fragmentation in aging. The critical protein for initiating the complex fission process is dynamin-related protein 1, Drp1, a GTPase that is recruited to fission sites and forms a large complex around the fission site that initiates the fission process [181,189,190,191,192]. Enhanced Drp1-dependent fission in midlife promotes a healthy lifespan in D. melanogaster [193]. It appears that the same signals that recruit PINK1 to damaged mitochondria also recruit Drp1 to the fission sites, namely mROS and ∆Ѱ collapse. This process is mediated by the phosphorylation of Drp1 by GSK3b (which is activated by ROS) [189], and also by phosphorylation of MFF, a receptor of DRP1, by AMPK (which is also activated by ROS) [140,145,146]. Additionally, PINK1, which is recruited to the mitochondria by the collapse of ∆Ѱ, also phosphorylates Drp1 [194]. The combination of enhanced mROS and ∆Ѱ collapse is indicative of mPTP activation, and there is direct evidence that mPTP enhances mitochondrial fission [195]. ROS was also shown to recruit PARKIN to OMM [196,197]. The recruitment of PINK1 to damaged mitochondria appears to depend on mPTP activation. The best known method for the initiation of PINK1 accumulation on OMM is by collapsing ∆Ѱ with uncoupler [198], which is known to induce the activation of mPTP [2]. Several studies showed that the activation of mPTP enhances mitophagy [199]. For example, Q deficiency, which was shown to activate mPTP, increases mitophagy/autophagy, and that effect was inhibited by cyclosporin, but in Atg5 knockout mice (which inhibits autophagy), Q deficiency resulted in apoptosis [200,201]. Additionally, the overexpression of CypD enhances mitophagy/autophagy [202]. PINK1 accumulation on OMM also depends on ANT [20] but it is not clear whether this effect of ANT on mitophagy depends on direct interactions or results from the dependence of mitophagy on mPTP. Because mitophagy inhibits extended mPTP activation in the cell by removing fragmented mitochondria with activated mPTP, it is an important antiaging mechanism. Life extension by calorie restriction or inhibition of the insulin/IGF1 pathway in C. elegans depends on mitophagy [203]. Similarly, life extension in C. elegans by urolithin A depends on mitophagy [204]. Mitophagy also retards aging by inhibiting the formation of the NPLR3 inflammasomes [205,206,207,208], which are also, apparently, induced by mPTP activation [14,164,209]. Figure 2 shows the fission/fusion and the mitophagy/autophagy processes that clear mPTP-activated mitochondrial fragments and restore a functional mitochondrial network.

Mitophagy was shown to be inhibited in aging [138,160,203,210,211,212] and this inhibition probably contributed to the enhancement of mPTP activity in aging. One reason mitophagy is inhibited in aging is the loss of cellular NAD^+^ in aged cells [89], which, as discussed above, also partially results from mPTP activation. Similarly, the inhibition of SIRT3 activity (which depends on NAD^+^) in aging also inhibits mitophagy [111,213]. Aging appears to accelerate fission [187], and since aging also inhibits fusion, through the loss of OPA1 [188], and inhibits mitophagy, the result of these three effects is the enhancement of mPTP-driven aging and eventual cell death.

Parkinson’s disease is a mitochondrion-dependent, aging-driven, neurodegenerative disease in which the death of dopaminergic neurons, particularly in the *substantia nigra*, leads to progressive movement disorders [214,215]. There are two main forms of Parkinson’s disease: familial and sporadic, and both depend strongly on age. An important driver of Parkinson’s disease is oxidative stress [216,217,218]. Major contributors to the sporadic form of the disease, in addition to aging, are exposure to pesticides and other toxins such as rotenone, paraquat and MPP+ that increase ROS production and activate mPTP [219,220,221,222,223,224,225,226]. The familial forms of Parkinson’s disease result from mutations in a number of proteins: mitochondrial proteins that participate in mitophagy, PINK1 and Parkin [186,227], LRRK2, a protein that participates in fission [228,229,230,231], several ATG proteins that participate in autophagy [212] and α-synuclein [231]. Thus, the majority of the familial forms of the disease result from mutations in proteins that participate in different stages of mitophagy/autophagy, indicating that the disruption of mitophagy is a major cause of the familial form of the disease [212]. It is not entirely clear how mutations or oxidative damage to α-synuclein result in disease [215,232,233]. However, it was reported that mutated, aggregated or oxidatively damaged α-synuclein activates mPTP, apparently by direct interaction with ATP synthase [158,234,235]. It is possible that the direct activation of mPTP by oxidatively damaged α-synuclein is the major route for the oxidative stress-induced activation of mPTP in Parkinson’s disease. Since the inhibition of mitophagy in most of the familial forms of Parkinson’s disease will result in the accumulation of fragmented mitochondria with activated mPTP, and the toxins that cause Parkinson’s disease induce the excess production of mROS that activates mPTP, it appears that the activation of mPTP is the major cause of cell death in Parkinson’s disease. The ROS-induced activation of mPTP in the electron transport inhibitor model (MPTP^+^) of Parkinson’s disease leads to activation of the NLRP3 inflammasome, resulting in the loss of dopaminergic neurons [164,205]. It is therefore increasingly evident that Parkinson’s disease, in all of its manifestations, is caused either by the inhibition of mitophagy (which fails to remove activated mPTP) [169,170,186,192,215,227,236] or by the excessive activation of mPTP [164,235], as summarized in Figure 3. Aging enhances the production of mROS and this can increase mPTP activation directly or through oxidative damage to α-synuclein. Moreover, aging also inhibits mitophagy which explains the strong dependence of Parkinson’s disease on aging. Dopaminergic neurons are more susceptible to mPTP activation than other cells because they are particularly common in aging, causing the overactivation of mPTP, thereby leading to cell death [237]. The overactivation of autophagy/mitophagy is a major factor in a variety of other neurodegenerative diseases [238].

## 5. Lifespan and Healthspan Extension Paradigms and mPTP

There is currently a great effort to discover drugs, nutritional supplements or lifestyle modifications that extend lifespan and healthspan [239]. Since current evidence suggests that mPTP activation accelerates aging and age-driven degenerative disease, it appears that mPTP itself could be a target for drugs that extend lifespan or retard aging-driven degenerative disease. Indeed, cyclosporine A was shown to protect against I/R damage and retard several age-driven diseases [14]. However, cyclosporine is a nonselective inhibitor of cyclophilins and is known to suppress the immune response, a fact that greatly limits its utility as an antiaging drug. Cyclosporine derivatives that are more specific for CypD do show more promise in this regard [240]. Nevertheless, to date, the effort to identify mPTP inhibitors that are clinically useful has not been successful. While this effort is ongoing, and may still result in useful drugs [1], it is possible that drugs that directly block mPTP would not have wide application because these drugs do not distinguish between short transient openings of mPTP, that are beneficial, and the long full activation of mPTP, which is damaging. We believe that what is needed is a drug, or other manipulations, that only inhibit damaging, long, full opening of mPTP and not the short, partial opening that can be beneficial. Recent data suggest that many lifestyle modifications, drugs and nutritional supplements that appear to extend lifespan and retard age-driven degenerative disease do indeed protect against the hyperactivation of mPTP in the context of aging and disease.

As was discussed above, autophagy is a major mechanism to retard aging and aging-driven degenerative disease, and Zhou et al. [27] demonstrated, in experiments with C. elegans mutants, that several major autophagy-dependent mechanisms of lifespan extension depend on mPTP inhibition. Hyperactivation of mPTP in these mutants (by overexpression of VDAC1) reverses the lifespan extension of these mutations. It is also clear that the induction of mitophagy, which eliminates fragmented mitochondria with activated mPTP, is a major contributor to the antiaging effect of autophagy.

Rapamycin, an mTORC inhibitor, extends lifespan and retards aging [241] and it is well established that rapamycin inhibition of mTORC1 activates autophagy [242,243,244]. These effects suggest the inhibition of mPTP activity. Indeed, rapamycin was shown to reverse (the mPTP-induced) mitochondrial fragmentation [242].

Melatonin is a pineal hormone that controls the circadian cycle and is known to have a protective effect against neurodegeneration, heart disease and cancer, which is mediated through the inhibition of mPTP [245,246,247,248,249]. It has been shown that melatonin is a potent inhibitor of mPTP in isolated mitoplasts [250]. However, the exact mechanism of inhibition is not clear. Highly significant is the observation that melatonin does not inhibit the transient (and beneficial) opening of mPTP [251]. It is therefore clear that the inhibition of mPTP by melatonin is indirect, and that melatonin only inhibits the damaging full opening of mPTP. This conclusion is also supported by the fact that melatonin is a widely used supplement, taken by millions of people, apparently without any deleterious effects.

Metformin is a widely used antidiabetic drug that has been shown to extend lifespan in animal models of aging, and to increase human healthspan [252,253,254]. Metformin was also reported to activate mitophagy [255]. It is known that metformin directly inhibits NADH dehydrogenase, and it was shown that this inhibition leads to the inhibition of mPTP and protection from I/R damage [256,257,258]. However, it is not known how the inhibition of NADH dehydrogenase translates into the inhibition of mPTP. Apparently, the enhancement of mROS production that results from the inhibition of NADH dehydrogenase activates autophagy by inhibiting mTORC1 [252,259] and by activation of AMPK [260,261]. Thus, metformin may be another example of a drug that inhibits only the aging-inducing full activation of mPTP, but does not inhibit the beneficial transient opening of mPTP.

Resveratrol, an antioxidant, is known to enhance healthspan [262]. Resveratrol was shown to enhance autophagy and mitophagy [263,264], and it appears that this effect also depends on the inhibition of mPTP [265,266,267,268]. Resveratrol protects against I/R damage in myocytes by the dephosphorylation of VDAC1, which inhibits mPTP [268]. Similarly, protection from ER stress by resveratrol depends on the inhibition of mPTP [269]. Resveratrol was also shown to protect against neurodegeneration by activating sirt1 which activates PGC1a that accelerates mitochondrial biogenesis (which replaces mPTP-damaged mitochondria with newly minted mitochondria) [270,271]. Resveratrol activation of sirt3 (which inhibits mPTP) was also demonstrated in several studies [271,272,273].

Spermidine is a known inducer of autophagy [263,274] and has been shown to be an effective antiaging agent [275,276,277]. Spermidine induces autophagy by inducing the synthesis of the autophagy transcription factor TFEB [278] through the AMPK–mTORC1–ULK1 pathway [274]. It was shown to provide cardioprotection and to extend life in mice through the activation of autophagy and mitophagy [276]. Spermine, a metabolite of spermidine, also has a cardioprotection effect [279]. Therefore, it is more than likely that there is also a direct effect of spermidine on mPTP since spermine and other polyamines have been shown to inhibit mPTP in isolated mitochondria [280,281,282].

Exercise and dietary restriction are two known lifestyle modifications that enhance healthspan. Exercise is a well-established lifestyle modification that retards aging and increases human healthspan [283]. Exercise enhances mitophagy and autophagy [284,285], which are associated with the inhibition of mPTP. It was shown that exercise training decreases susceptibility to Ca^2+^-induced mPTP opening in heart mitochondria [286]. It was also demonstrated that endurance exercise in hyperglycemic rats decreases susceptibility to mPTP opening in isolated heart mitochondria [287]. Similarly, exercise protects against the enhanced mPTP opening in heart mitochondria of rats treated with doxorubicin [288]. Dietary restrictions have been shown to increase lifespan and healthspan in all animal models of aging (e.g., yeast, C. elegans, Drosophila, mouse) [239,289,290,291]. It is evident that the nutrient-sensing mTOR and the insulin/IGF1 pathways mediate the effect of dietary restriction on aging [149,291]. However, the mechanism(s) that lead from these signals to life extension are not entirely clear. Apparently, induction of autophagy, mitophagy, mitochondrial metabolism modification or antioxidant response could be the critical element in various paradigms of dietary restriction [292]. Nevertheless, it is also evident that these pathways may all result, directly or indirectly, in the inhibition of mPTP. Several studies demonstrated that dietary restriction prevents mPTP opening in liver and brain mitochondria [293,294,295], but not in the skeletal muscle or heart [296]. Zhou et al. [27] showed that the increased lifespan of the calorie-restricted eat-2 C. elegans mutant is dependent on the inhibition of mPTP, similar to other autophagy-dependent life extension paradigms. Dietary restriction in humans was shown to reduce oxidative stress [297], and since oxidative stress is both a major cause of enhanced mPTP activity, and an outcome of enhanced mPTP activation [14], it is likely that mPTP activity is reduced in ageing humans subjected to dietary restrictions.

## 6. Conclusions

Almost half a century ago, it was first proposed in the mitochondrial free radical theory of aging that mitochondrial reactive oxygen species, mROS, are the major cause of aging and thus determine the lifespan of animals and humans. Four decades ago, the mitochondrial permeability transition pore, mPTP, was first discovered, and two decades ago, it was first shown that mPTP activity is enhanced in aging. Over the last two decades, extensive research on aging and aging-driven degenerative diseases, and on the many pathways that control mPTP activation, have brought these apparently unrelated fields together into an emerging understanding of the connection between these phenomena. While the mitochondrial free radical theory of aging first appeared to be challenged by the discovery that mROS signaling actually protects against aging and disease, there is now a better understanding of the role of mROS signaling, driven by a modest increase in mROS production, in activating protective mechanisms against the damaging effect of excess mROS production. Moreover, it is becoming clear that mPTP plays a critical role both in mROS signaling, by partial, short openings of the pore that release small amount of mROS, and in mROS-induced aging, by the full, extended opening of mPTP that releases large amounts of mROS and NAD+ that damage the cell and accelerate aging and aging-dependent diseases. Recent studies show how the complex control of mPTP activity can play a critical role in both the mechanisms that protect the cell from aging and disease, and in the mechanisms that accelerate aging and drive the aging-dependent degenerative diseases. The major pathways that control the activity of mPTP are summarized schematically in Figure 1. In particular, the recent discovery that mPTP activity determines whether autophagy/mitophagy protects from aging and disease or accelerates cell aging and death greatly clarifies the decisive role of mPTP in aging and disease and can guide the discovery of new drugs and lifestyle modification that enhance healthspan and lifespan.

## Figures and Tables

**Figure 1 cells-10-00079-f001:**
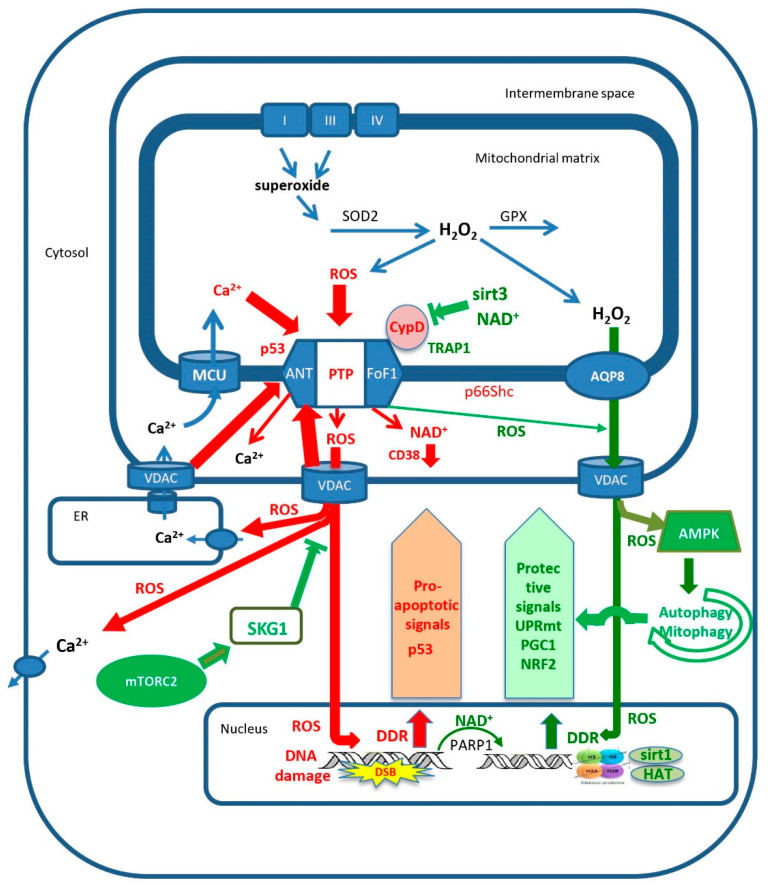
The control of mitochondrial permeability transition pore (mPTP) activity determines the progression of aging. Full opening of mPTP, which is activated by excess mitochondrial calcium loading and/or excess mROS production, releases large amounts of calcium, NAD+ and mROS from the mitochondrial matrix. The released NAD+ is hydrolyzed by CD38, and the loss of NAD+ enhances the progression of aging by inhibiting sirtuins and PARP1. The release of large amounts of mROS causes nuclear DNA damage which enhances apoptotic signaling, such as P53 and p66Shc, further enhancing the activation of mPTP. Additionally, mROS causes oxidative damage to calcium transporters, which enhances excess loading of mitochondrial calcium that, together with the excess release of mROS, activates additional full opening of adjacent mPTP sites. In contrast, modest increases in calcium and/or mROS trigger partial short opening of mPTP, releasing small amounts of mROS that, together with slow diffusion of mROS through AQP8, activate mitochondrial protection mechanisms such as autophagy/mitophagy, UPRmt, NRF2 and PGC1. Additionally, mTORC2 activates SGK1 that inhibits the voltage-dependent anion channel (VDAC) from activating mPTP, enabling autophagy/mitophagy to protect the cell from the progression of aging. Excessive activation of mPTP turns autophagy/mitophagy into a destructive process that leads to cell death. See text for further details.

**Figure 2 cells-10-00079-f002:**
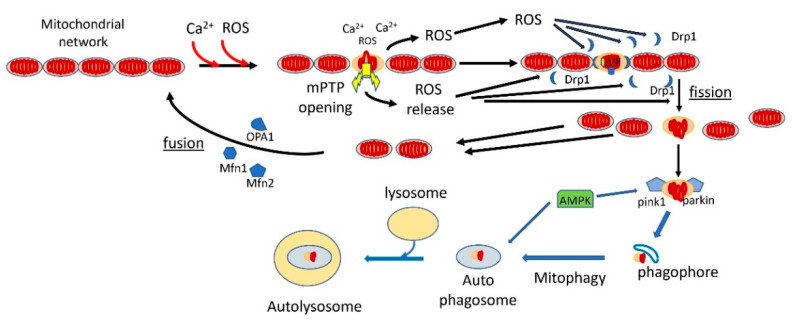
Mitophagy retards aging by clearing mitochondrial fragments with fully activated mPTP. In young normal cells, the mitochondria are connected in a mitochondrial network. With aging, increased mROS production and mitochondrial calcium overloading fully activate mPTP in some mitochondria. The mROS released by mPTP induces mitochondrial fission by recruiting dynamin-related protein 1 (Drp1) to contact sites between the mitochondria. The depolarized, mPTP-damaged fragments recruit PINK1 and Parkin, which leads to ubiquitination of the mPTP-damaged fragments, labeling them for mitophagy. The mROS produced by mPTP activation also activate AMPK, which enhances autophagy/mitophagy. The damaged, mitophagy-labeled fragments are then engulfed by the phagosome, which progresses into autophagosomes. These are taken out by lysosomes, where the damaged mitochondrial fragments are degraded. The undamaged fragments recruit OPA1 and mitofusins and are fused back, together with newly synthesized mitochondria, into the mitochondrial network.

**Figure 3 cells-10-00079-f003:**
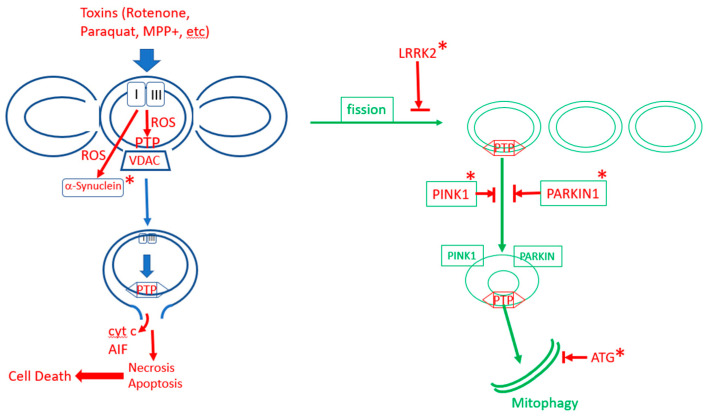
The role of mPTP in Parkinson’s disease. In sporadic Parkinson’s disease, excess production of ROS in dopaminergic neurons, due to aging and/or toxins, activates excess opening of mPTP, leading to cell death. Mitophagy protect neurons by removing mPTP-activated mitochondrial fragments. In familial Parkinson’s disease, mitophagy is inhibited by mutations (labaled by “*”) in enzymes that promote mitophagy (LRRK2, PINK1, PARKIN, various autophagy-related proteins (ATGs)), or mPTP is enhanced by mutations in α-synuclein.

## Data Availability

Not applicable.

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
