# Peer review of "The Mitochondrial Permeability Transition: Nexus of Aging, Disease and Longevity"

_cells, 2021, doi:10.3390/cells10010079_

Round 1

Reviewer 1 Report

This is a well-written and well-organized review article that discusses the role of Mitochondrial Permeability Transition in Aging, Disease and Longevity.  

Although some specific aspects on this topic were previously covered by other reviews, the paper nevertheless contains significant information/connection of current interest. The Authors are experts in the field and were able to simplify complicated issues, with the aim of providing an integrated view of the connections between mPTP and aging vs longevity, health vs disease. 

I have no major revision to ask. The following suggestions are aimed at improving the readability of the paper.

  • Although it could seem redundant, I recommend introducing the mitochondrial membrane localization of mPTP in the initial description of this complex.
  • There are some mistakes in the reference section (60, 65 and 73 are erroneously inserted) that change the subsequent numerical order. From then onwards, the numbers in the text and the numbers in the reference section do not pair up. Please modify. 
  • A graphical representation of the cell death mechanisms in Parkinson’s disease would clarify the discussion of the data reported in the last part of paragraph 4.
  • It would be helpful to conclude the review article with a scheme summarising the most relevant pathways involved in the negative or positive regulation of mPTP in aging vs longevity and health vs disease. It would also be advisable to include an indication of the putative targets of the drugs/lifestyle modifications that potentially protect the cells from aging and disease.

Author Response

we accepted the suggestion to add a figure that clarify the role of mPTP in Parkinson’s disease. We added Figure 3, which show schematically the relationship between Parkinson’s disease, mPTP activation and mitophagy (page 9, lines 366-372). Also, as suggested we slightly expanded the description of the complex in the introduction (Page1, line 32) and added a sentence to the conclusion section, directing the reader to Figure 1, that summarizes schematically the regulation of mPTP in the context of aging and disease (page 12, lines 476-477). We corrected the reference list (our reference list was scrambled by the editorial office automatic formatting of our original manuscript. We corrected it, as much as Cells formatting let us, but we suspect that the automatic formatting may scramble it again). We declined to extend the discussion to the putative targets of the anti-aging drugs/lifestyle modification beyond the studies, which we discuss, that show that they inhibit mPTP and/or enhance autophagy/mitophagy. In most cases, there is no consensus about the direct targets and discussing the hundreds of studies devoted to these issues are best left to separate reviews on these drugs/lifestyle modifications.

Reviewer 2 Report

In this review, the authors have comprehensively summarized recent progress in mPTP and its relationship with aging and age-related disease (PD). The writing is precise and clear. The schematic figures were well illustrated and are helpful for the reader's understanding of the topic.

Indeed, although mPTP is a critical mitochondrial event, there are so many unknowns. In addition to the unresolved molecular identity of mPTP, the importance of the physiological functions of this mitochondrial pore has emerged. The discussion about the roles of mPTP in neurophysiology is highly suggested. 

Author Response

In response to the suggestion to discuss the role of mPTP in neurophysiology, we added a sentence to the introduction on the proposed role of mPTP-induced superoxide flashes in cortical neurons progenitor differentiation (page 2, lines 49-50, reference 12). However, we declined to extend our review for full discussion of this subject, which is not directly related to the subject of our review. We do discuss, in length, the role of mPTP in neuronal pathology which is the subject of our review.

Reviewer 3 Report

This is a very well written and balanced review on the pathways that activate or inhibit the mitochondrial transition pore in the context of aging and degenerative diseases. The single point that I would have like to see a bit more extended relates to the composition of the pore. It is explained briefly in the introduction, but it could be extended to mention whether the transition pore is opened in respiratory deficient cells, cell devoid of mtDNA, and the implications for the composition and regulation of the pore. Also, the authors could provide their view regarding whether a single pore entity exists or if the permeability transition could rely on multiple pores of diverse composition.

Author Response

We followed the reviewer advice and added to the introduction a reference and discussion of the fact that cells devoid of mtDNA still exhibit mPTP activity (Page 2, lines 64-64, reference 19). We think that within the framework of the introduction, our summary of the recent work on the composition of mPTP, and the many recent references on this subject that we provide, is sufficient for the purpose of this review.